# Atmospheric conditions of extreme precipitation events in western Turkey for the period 2006-2015

5    **Bulent Oktay Akkoyunlu[1], Hakki Baltaci[2,*], Mete Tayanc[3]**

1: Marmara University, Department of Physics, Istanbul, Turkey
2: Turkish State Meteorological Service, Regional Weather Forecast and Early Warning Center, Istanbul, Turkey
3: Marmara University, Department of Environmental Engineering, Istanbul, Turkey

* Correspondense to: baltacihakki@gmail.com

**Abstract**

This paper investigates the precipitation types and background physical mechanisms of

15    extreme precipitations events (EPEs) over western Turkey during the period 2006-2015.

The EPEs are described as the precipitation values above the 90[th] percentile obtained

from the hourly precipitation dataset having high spatial resolution. Precipitation types

associated with EPEs are identified by using radar outputs and Lamb Weather Type

(LWT) approach. It is found that EPEs occurred more frequently in the Marmara and

20    Aegean regions during autumn and winter months. In Marmara, mainly 21%, 17% and

15% of total autumn EPEs show convective, cyclonic, and sea-effect precipitation

characteristics, respectively. While convective EPEs are seen more common in the

southern portions, cyclonic and sea effect originated EPEs mainly affect the southwest

and northeastern parts of Marmara. Among these three precipitation types, convective

25    mechanisms generally produce more intense daily precipitation (66.1 mm in average) in

the Marmara region under the proper synoptic conditions (high-pressure center over

Balkan Peninsula and low-pressure center over eastern Mediterranean). Based on the

hourly observations, convective types of extreme precipitations (EPs) show two peak

values during afternoon and evening times of the day and are linked to diurnal heating. In terms of Aegean region, cyclonic originated EPs, which include 65% of the total winter EPEs, are more common in the whole territory and reach to its peak value during the first hours of the day.

## 1. Introduction

The occurrence of extreme precipitation events (EPEs) and background physical mechanisms triggering these episodes become a fundamental issue in the last decade due to its great impacts on agriculture, health, energy, and tourism. From this perspective, many researchers firstly identified the EPEs by applying fixed (e.g. Brooks and Stensrud 2000; Ralph and Dettinger 2012; Hitchens et al. 2012, 2013) or percentile based precipitation thresholds (e.g. Piccarreta et al., 2013; Krichak et al., 2014) to the daily precipitation. In the later studies, the main atmospheric systems that cause extreme precipitations (EPs) are investigated in detail by focusing the role of large-scale (e.g., Madden-Julian Oscillation (MJO), ENSO, PDO) (Jones, 2000; Higgins et al. 2000; Deflorio et al. 2013). or the synoptic scale circulations for the selected regions in US (Schumacher and Johnson 2006; Warner et al. 2012; Moore et al. 2015). Afterwards, the characteristics of the EPEs are defined by using radar (Moore et al. 2015), outgoing longwave radiation (Carvalho et al., 2002) or horizontal temperature advection data (Milrad et al., 2010).

Owing to the spatial complexity, rugged topography, and land-sea interactions of the Mediterranean Basin, many devastating flash floods occurred in the various part of the region in the last decade. Therefore, researchers have analyzed the atmospheric conditions that cause these extraordinary events by focusing on the selected flood days (e.g., Ferretti et al. 2000; Nuissier et al. 2008; Pastor et al. 2010). Only a few

researchers analyzed the climatological and general synoptic behaviors of the EPEs for this large territory (Ricard et al. 2011; Reale and Lionello 2013).

Turkey is located at the east Mediterranean and EPEs there, in general, cause sudden flash floods resulting with deaths and economic losses in infrastructure and agriculture. As a result of the EPEs in the last decade, numerous flash floods and landslides occurred in some particular regions of Turkey. During September 2009, Ayamama creek in Istanbul (NW of Turkey, most populated city in Europe) was overflowed as the consequence of the dense daily precipitation episodes, which produced more than 250 mm rainfall over 3-day, and 32 people died together with millions of dollars of economic losses (Kömüşçü and Çelik 2013). During 9 October 2011, 238 mm rainfall total was measured during an 6 hour time-period at the province of Antalya (south of Turkey) and damaged the infrastructure of the tourism center of the country (Demirtaş, 2016). During August 2015, torrential rainfalls ended up with a devestating landslide in Hopa district (NE Turkey, sloppy domain of the country) and 11 people died during this natural hazard (Baltaci, 2017). Turkey and its sub-basins are mainly influenced by these EPEs in all seasons in the variety of the atmospheric conditions such as baroclinic waves and cyclones, mesoscale convective systems, land-sea interactions and orographic forcing.

In literature, numerous studies investigated the influence of large-scale circulation patterns or synoptic weather types on precipitation mechanism over Turkey and its sub-regions (Karabörk and Kahya 2003; Karabörk et al. 2005; Unal et al. 2012; Baltacı et al. 2015, 2017). Only a limited number of these studies explored the atmospheric conditions that caused extreme precipitation over Turkey for a set of selected episodes (Kömüşçü et al. 1998; Kömüşçü and Çelik 2013; Demirtaş 2016).

Although a number of prior studies have focused on the synoptic characteristics of the EPEs ending up with life or economic losses over Turkey, environmental characteristics of these EPEs and underlying causes were not studied in detail. To overcome this deficit, we identified the types of the EPs, which are taken from ten-year (2006-2015) high-resolution precipitation datasets, by using the Lamb Weather Type (LWT) approach (Fig. 1a) and radar outputs in the western Turkey. Therefore, the goal of this study is to document the spatio-temporal and environmental characteristics of the EPEs, and investigate the synoptic-scale patterns associated with EPEs.

In Section 2, description of the precipitation characteristics of EPEs, along with the data and methods used, are described. Results of the EPEs and related discussion are presented in Section 3. The last part, Section 4, is devoted to the summary and conclusions.

## 2. Data and Methodology

### 2.1 Precipitation dataset

Values of meteorological parameters in Turkey had been recorded manually from the late 1920s to the beginning of the 21$^{st}$ century. After the year 2003, starting from the western regions, existing meteorological stations were replaced by automatic ones (Automatic Weather Observing Systems, AWOS) and also virgin land was covered with AWOS stations by the support obtained from large projects. These projects can be explained in four parts as follows:

1) AWOS 206: Excessive rainfalls on May 21-25, 1998, which also triggered landslides, resulted in many flash floods over the western Black Sea region of Turkey. In order to eliminate damages originated from floods, TEFER project (Turkey Emergency Flood and Earthquake Recovery) was introduced that was

financially supported by International Public Works and Development Bank (World Bank) by a fund of 369 million dollars, to strength the emergency early warning systems in the west Turkey. As a part of this project, 206 AWOS stations as a total have been started to be operational during 2003 and 2004 (red points in Fig. 1a). With this project, 120 classical meteorology stations were replaced by the new automated ones and additional 86 AWOS stations were installed into new areas.

2) AWOS 151: After the setup of 206 AWOS, extra 151 AWOS stations were installed in the central and eastern parts of Turkey during 2009 (brown triangles in Fig. 1a). Out of 151, 120 manual meteorology stations were transformed into new tech ones and the remainder 31 were located in the maiden areas of the country.

3) AWOS 246 and 350: To expand the spatial density of the meteorological stations, 246 and 350 AWOS stations have been started to be used in the following years. By the 246 stations, new districts were aimed to be meteorologically covered, which did not have any active meteorological stations before (blue stars in Fig. 1b). Later, due to high topographical differences of the country, 350 new automated meteorology stations were mainly located at the higher elevation points and started to operate since 2016 (black squares in Fig. 1b).

In our study, for the first time, we aimed to obtain the atmospheric conditions of EPEs in Turkey with high resolution and coverage. Therefore, we have chosen long-term hourly precipitation dataset of AWOS stations. For this reason, hourly precipitation records of 206 AWOS stations were selected for the investigation for the environmental

characteristics of EPEs. Firstly, daily total precipitation amounts (00-24 UTC) were calculated from hourly precipitation records. Quality controls of data were done by RCLIMDEX method which was explained by Zhang and Yang (2004) and Baltacı et al. (2018). The years having more than 10 % missing data days and stations that are subjected to relocation were eliminated from the study. As a consequence of the quality control and assurance of precipitation data in the period 2006-2015, we selected 97 stations densely located in the west Turkey (Fig. 1c). From 97, 51 stations are found to be located in the Marmara (NW Turkey, pink points), and 46 to be located in the Aegean (W Turkey, light brown points) regions of Turkey.

## 2.1.1 Climatic characteristics of Marmara and Aegean regions

The Marmara Region is located in the northwest of Turkey, between latitudes 29°N and 32°N and longitudes 38°E and 42°E and covers an area of 67000 km$^2$. Marmara has different climatic characteristics in itself. While inland areas have temperate continental climate, milder climate of places on the Black Sea coast resembles more of an oceanic climate, typical to other areas of Turkish Black Sea coast. The coasts in Marmara and Aegean parts have Mediterranean (Med) climate and this region is second smallest Turkish region in size after Southeastern Anatolia. Only south and east parts of the region are more mountainous.

In terms of Aegean, this region has a Med climate with mean annual precipitation changing from 450 to 1200 mm yr$^{-1}$ (Asikoglu and Benzeden 2014). Although climatic behavior of the Aegean is similar to the Med climate, it is shown obvious differences in landscape. Unlike the more parallel mountains found along the Med, Aegean mountains often cut directly into the sea.

## 2.2 Radar data over Turkey

First meteorology radar over Turkey was installed in Ankara for nowcasting purposes during the year of 2000 (Fig. 2). Afterwards, Istanbul, Zonguldak, and Balikesir radars were installed during 2003 by the TEFER project. Later, to detect EPEs that can be effective over Mediterranean and Black Sea, another six C band radars were setup to Izmir, Mugla, Antalya, Hatay, Samsun, and Trabzon cities during 2007. Due to the forecast difficulties of convective precipitation by the numerical weather prediction models, another four C band radars (Bursa, Afyon, Karaman, and Gaziantep) were located in the inner parts of the country during 2013. From the network of these 14 radar stations, we used Istanbul, Zonguldak, Ankara and Balikesir radar 8-min PPI (Plan position indicator), and Max (Maximum) products, which are provided by Turkish State Meteorological Service (TSMS), to check the characteristics of EPEs (cyclonic, convective or sea-effect originated) in combination with Lamb Weather Type (LWT) classification (e.g., Hellström 2003; Burt and Ferranti 2012; Moore et al. 2015). Additionally, data of the other radars, Izmir and Mugla were analyzed in the study from the beginning of 2007, and data of Bursa and Afyon from the year of 2013.

## 2.3 Lamb Weather Type (LWT) methodology

The subjective version of the Lamb's work (Lamb 1972) was firstly developed as an objective version by Jenkinson and Collison (1977) and refined by Jones et al. (1993) to indicate the circulation types (CTs) influencing British Isles. According to the objective methodology, vorticity and directions of the geostrophic flows are calculated using sea level pressure (SLP) fields over a predetermined central point. As a consequence of the six parameters and certain thresholds for the defined region, totally 27 different CTs were defined (16 hybrid, 8 directional, cyclonic, anticyclonic, and unclassified types). In this study, we used daily mean SLP values on 16 grid points

(between 5°W-55°E and 30°N-60°N, Figure 1a), centred over Marmara Region and separated by 5° from each other. The six parameters, namely the westerly flow (WF), southerly flow (SF), resultant flow (FF), westerly shear vorticity (WSV), southerly shear vorticity (SSV), are computed as follows:

$$WF = \left[ \frac{1}{2}(p_{12} + p_{13}) - \frac{1}{2}(p_4 + p_5) \right] \tag{1}$$

$$SF = 1.305 \left[ \frac{1}{4}(p_5 + 2 * p_9 + p_{13}) - \frac{1}{4}(p_4 + 2 * p_8 + p_{12}) \right] \tag{2}$$

$$FF = \left( WF^2 + SF^2 \right)^{0.5} \tag{3}$$

$$WSV = 1.12 * \left[ \frac{1}{2}(p_{15} + p_{16}) - \frac{1}{2}(p_8 + p_9) \right] - 0.91 * \left[ \frac{1}{2}(p_8 + p_9) - \frac{1}{2}(p_1 + p_2) \right] \tag{4}$$

$$SSV = 0.85 * \left[ \begin{array}{c} \frac{1}{4}(p_6 + 2 * p_{10} + p_{14}) - \frac{1}{4}(p_5 + 2 * p_9 + p_{13}) \\ -\frac{1}{4}(p_4 + 2 * p_8 + p_{12}) + \frac{1}{4}(p_3 + 2 * p_7 + p_{11}) \end{array} \right] \tag{5}$$

$$Z = WSV + SSV \tag{6}$$

where $p_i$ is the daily mean SLP at grid point $i$ (Figure 1a). Finally, classification of CTs is done according to the following criteria:

- Directional types (N, NE, E, SE, S, SW, W, NW) is found by tan$^{-1}$ (WF/SF), adding 180° to the final value if WF is positive. 45° is allocated for each sector.

- If |Z|<FF, CT is one of the eight pure directional types listed above

- If |Z|>2FF, the CT is either Cyclonic or Anticyclonic

- If FF<|Z|<2FF, the CT is one of the 16 hybrid types: a combination of directional and vorticity types.

- If |Z| or FF<6, then the CT is 'unclassified'

For the Aegean region, the equations based on six circulation parameters for Marmara were also recalculated by using different 16 grid points (centred over Aegean) and coefficients (due to latitudinal difference).

## 2.4 Identification of EPEs and precipitation characteristics

An extreme precipitation event is generally defined as a daily amount exceeding a certain threshold (e.g. Brooks and Stensrud 2000; Ralph and Dettinger 2012; Hitchens et al. 2012, 2013). For example, Karl et al. (1996) used 50.8 mm to define extreme precipitation events for the United States. For our country, this and other threshold limits was not suitable because of the large topographical difference and irregularity of the precipitation distribution. For this reason, similar to the previous studies (e.g. Jones, 2000; Zhang et al., 2001; Piccarreta et al., 2013; Krichak et al., 2014), we chosen a methodology that defines the threshold levels of each station according to its own precipitation characteristics. Thus, this relative technique is based on considering the largest 10% of the daily precipitation amounts of each station separately as its own extreme. Then, the annual contribution of EPs for each station is determined by a standardized total that is the division of the cumulative totals of EPs for each station by 10.

We defined EPE types by using the LWT technique and the radar outputs. For LWT technique, basic air flows coming from the sea and generating extreme precipitation, defined as sea-effect precipitation. The cyclones, which generate severe precipitation over a defined region, are also characterized with cyclonic EPEs. We identified the convective EPEs as the precipitation bands coming from the terrestrial areas. As a result of these definitions, EPEs having northerly (N) and northeasterly (NE) CTs for the Marmara region were considered as having Black-Sea effect. For the sea-

effect EPEs over Aegean, westerly (W) and southwesterly (SW) CTs were selected. For the convective EPEs, easterly (E), southeasterly (SE), and southerly (S) CTs for Marmara and E and SE types for Aegean were chosen. In terms of cyclonic (C) EPEs, low-pressure center over Marmara and Aegean was selected as cyclonic CT in accordance to LWT methodology.

The physical mechanisms behind the EPs were investigated by using NCEP/NCAR Reanalysis products (Kalnay et al. 1996). For this purpose, sea level pressure (SLP) and temperature data of 850-hPa were examined on a 2.5° X 2.5° grid resolution of the reanalysis data. For the sea surface temperature (SST) distribution over the neighboring sea areas around Turkey, NOAA High Resolution SST data provided by the NOAA/OAR/ESRL PSD, Boulder, CO, USA, from their website at http://www.esrl.noaa.gov/psd, (Reynolds et al. 2007) were used in the study.

## 3. Results and Discussion

### 3.1 Spatial variation of EPEs in the west Turkey

For the first time, different daily precipitation threshold limits of 97 stations were constructed from a 10-year dataset (Fig. 3a). According to the results, highest daily precipitation rates exceeding 100 mm are observed on the southern Aegean region where it can be classified as 'rich' in terms of extreme amounts of precipitation. This suggests that if the daily precipitation amount of a station located in the south Aegean exceed this limit, that day is recorded as an EPE for that station. Daily precipitation threshold ranging from 60 to 100 mm is shown to be mainly located on the coastal regions of the west Turkey. When one move towards interior continental areas, daily EP threshold decrease from 60 to 20 mm level. The lowest limits are observed in the semi-arid continental areas of the Aegean and Marmara region as having threshold value

lower than 40 mm, as illustrated with blue color in Fig. 3a and can be classified as 'poor' in terms of extreme amounts of precipitation.

The annual contribution of EPs for each station (cumulative totals of EPs for each station divided by 10) is shown in Fig. 3b. We observe that the largest normalized annual amounts of EPEs is located mainly on the southwest of Aegean, middle-south and northeast of the Marmara region with values larger than 60 mm. It is interesting to see that the interior continental areas of the Aegean and Marmara region that was characterized as poor in terms of extreme amounts of precipitation (Fig. 3a), now exhibit a better picture in their normalized value as generally having a better value between 40-60 mm. The reason of this can be the convective precipitation, generating intensified rain that can accumulate higher amounts of precipitation during a single rainstorm. On the other hand, western regions of Marmara that exhibited considerably larger threshold value with precipitation totals larger than 60 mm (Fig. 3a), show a worse image with the normalized values as having precipitation values between 40 to 60 mm.

As an example on October 28, 2010 intense daily rainfalls and associated many flash floods occurred on the western parts of Turkey. As a result of this extraordinary event, daily precipitation amounts exceeded 70 mm in the Bandirma province (south seaside station of Marmara Sea, in Fig. 4). During this day, daily precipitation totals exceeding 50 mm is shown in yellow color on Fig. 4 that is extending from coastal Aegean region towards Marmara as an enlarging region and reaching up to Black Sea passing over Gulf of Izmit and Silivri. This squall line affected majority of the Marmara region and to a lesser extends the Aegean region. However, many stations located

outside of this critical yellow region also had rainfall totals above their extreme daily precipitation limits.

Seasonal distribution of the EPE frequencies can provide important information to understand the physical mechanisms forcing these extreme events. For this reason, we analyzed total counts of EPEs for four seasons and the results are depicted in Fig. 5. It can be stated from Fig. 5 that winter (DJF) and autumn (SON) are more significant than the other seasons (Figs. 5a, d). During winter, two cores over Aegean result in more than 6 extreme precipitation days (Fig. 5a). Spring is mainly characterized as having EPEs between 2 to 4 days on the eastern portions of the Aegean region (Fig. 5b). During summer, highest count of the EPEs with 3 days is shown to be located over the Black Sea effected areas of the Marmara region (Fig. 5c). Seasonally, second highest frequency of EPEs can be found in the autumn. In this season, an area extending from northeast to south of Marmara receive a frequency considerably higher than 6 days (Fig. 5d). From this point of view, a detailed analysis of the atmospheric systems generating EPEs and effecting Aegean region mainly during winter and Marmara region during autumn becomes important. Next section is focusing on this aim.

## 3.2 Regional features of the seasonal EPEs

In this section, we carried out frequency analysis of the seasonal EP events for Marmara region and documented the results in Table 1. Marmara basin seems to be the most sensitive to these occasional precipitation events during the autumn season. It is clear in the Table 1 that 53% of the EPEs (sea-effect, cyclonic, and convective originated) occurred a total of 43 days in this season, and followed by 27, 15, and 22 days in the summer, spring and winter months, respectively. During autumn,

convective, cyclonic and Black Sea-affected types of EPs are most influential over Marmara with the percentages of 21%, 17% and 15%, respectively.

At Aegean region that is represented by a total of 46 stations, EPs are more frequent during the winter months and totally 35 different winter days ended up with cyclonic EPs during the 10-yr period (Table 2). It can be seen from the table that cyclonic EPEs represent 61% of the winter–time extreme precipitation events belonging to the region. Second highest frequency belongs to the autumn with a value of 43% corresponding to 28 events.

## 3.2.1 Precipitation characteristics of EPEs over Marmara with its background synoptic-scale atmospheric conditions

As discussed in the previous section, we mainly focused on the months of autumn to analyze the spatial distribution of daily mean precipitation, to determine the counts of station based EPEs in the Marmara region and investigate the synoptic-scale atmospheric conditions responsible from the development of these extreme precipitation events. In this respect, 2006-2015 period autumn mean precipitation values, counts of EPEs and their associated average weather maps are illustrated in Figure 6.

During cyclonic CTs, highest daily mean precipitation amounts exceeding 8 mm are shown to exist on the southwestern parts of the region. Similarly, the count of EPEs is higher on this portion of the Marmara region (Fig. 6a). When the synoptic composite maps are analyzed, one can see the low-pressure center that probably came from west (Karaca et al. 2000) and located over Aegean Sea and west of Marmara. Sea surface temperature varies between 19 and 20 °C and temperature in the low level of the atmosphere (approx.. 1.5 km high from the ground) is shown to be between 7.5 and 10 °C (Fig. 6b).

During NE types, north and northeast part of Marmara gets higher daily mean precipitation amounts (between 6 and 8 mm in Fig. 6c). Totally 28 extreme precipitation cases in the northeastern stations exceed their threshold levels at this part of the region. It is shown from the previous studies that primary factor for the formation and intensity of sea-effect precipitation is known to be the temperature difference between sea surface and the air at 850 hPa level (Holroyd, 1971; Niziol, 1987; Steenburgh et al., 2000). If the SST-$T_{850}$ difference becomes higher, the chance of precipitation increases due to higher convective instability. Millan et al. (1995) argued that enhanced evaporation resulting from temperature differences between European continental air and the relative warm Mediterranean Sea in fall can become a key factor in determining the onset of precipitation. Pastor et al. (2015) have shown that regions of high heat/moisture air-sea exchange over the Mediterranean Basin are prone to enhancing convection, leading to torrential rain. At the later study, Baltacı et al. (2015) mainly emphasized that 13 °C temperature difference between the sea surface and the 850-hPa level can cause above-normal precipitation records at the northeast of Marmara. Our results indicate that as a consequence of the combination of a high-pressure center (HPC) located over eastern Europe and a low-pressure center (LPC) over southern Turkey, strong northeasterly flows can be generated owing to high pressure gradient force bringing significant amounts of moisture from the relatively warm Black Sea (21 °C) to the northeast of Marmara. The temperature difference between SST and 850-hPa level exceeds 13 °C threshold level and this increase the strength of instability conditions (Fig. 6d).

As explained by Ricard et al. (2012), the orographical properties of the area induce mesoscale convergence and lift of the low-level conditionally unstable flow.

Most active regions for deep convection in Mediterranean Basin (MB) are the Alps, the western Croatian coast, the south of France and the wider area of Tunisia (Dayan et al., 2015). Alhammoud et al. (2014) found a maximum frequency of deep convection over MB in September-October and a minimum one in June and July. Similar to the previous studies for MB (e.g. Funatsu et al., 2009; Melani et al., 2013; Alhammoud et al., 2014), convective EP events over Marmara are mostly shown during autumn season. Although daily mean precipitation amounts are lower (between 0 and 2 mm) in the convective (E) type, extreme cases are seen more common in the southern part of the region (Fig. 6e). Mountainous area (Mt. Uludag over 2500 m high)) of Marmara is located in this part and due to the interaction between HPC over eastern Europe and LPC over western Turkey, strong easterly flows coming from flat land areas meet with highland barriers producing higher amounts of orographic enforced convective EPs, if the atmospheric condition such as temperature exchange (will be explained in Section 3.6) is suitable (Fig. 6f).

As an example, convective activity in southern Marmara started afternoon times on 28 September 2015 (Fig. 7). Due to the movements of the single cells to the easterly directions their spatial area expanded. When the cells met with orographic barrier over Bursa (Mt. Uludag, red star in Fig. 7) quasi-stationary conditions developed the convective instability. As a result, extreme precipitation amounts were recorded in the western part of the mountainside in a very short time.

## 3.2.2 Precipitation characteristics of EPEs over Aegean with its background synoptic-scale atmospheric conditions

As mentioned above, winter months are more important for extreme precipitation events over Aegean. As previously explained by Ulbrich and Christoph

(1999), high-pressure conditions in southeastern Europe tend to divert the Mediterranean storm track southwards, resulting in increased precipitation in the eastern Mediterranean. As a consequence the positioning on cyclones over Aegean in this season, more daily precipitation amounts occur in the southern corner of the region (above 14 mm) and we observe higher EP cases close to the coastal stations (Fig. 8a). In addition, during appropriate synoptic conditions, cyclonic activity can result in intense rainstorms at the majority of the stations of Aegean region, especially at those located in the south. When compared with cyclonic CT over Marmara, more deepened LPC is located over Aegean Sea. In this case, cold air aloft coming from north can meet with the relatively warm Aegean Sea and the convergence of warm air above the cold air can generate cyclogenesis which can result in heavy precipitation (Fig. 8b).

## 3.3 Interannual and hourly variation of EPEs

Annual distribution of the total counts of EP days together with their precipitation characteristics was also analyzed for Marmara and Aegean regions during autumn and winter months and depicted in Fig. 9. In terms of Marmara, cyclonic CTs were more active in 2009, 2011 and 2013 years (Fig. 9a). On the other hand, highest counts of Black Sea effected EPEs happened during 2008 and 2015 years. It is known that 2010 was a wet year for Marmara and dense daily rainfall amounts were generated by convective activity. Annual distribution of the cyclonic EPs in winter indicates that the Aegean region had been under the influence of mid-latitude cyclones during the years 2009 and 2010 (Fig. 9b). As previously indicated by Türkeş and Erlat (2003) that negative relationships were found between NAO indices and Turkish precipitation series. Statistically significant changes in the precipitation amounts during the extreme NAO phases are more apparent in the west and mid Turkey. Hence, one reason of this

cyclonic EPs could be the negative phase of NAO pattern, where warm and moist air over the Mediterranean Sea can be transferred to the Aegean region by strong westerly or southwesterly flow.

Peak times of extreme precipitation in a day can give us important information about understanding the possible causes triggering this event. For this reason, we investigated hourly behavior of the mean EPs for the Marmara and Aegean regions belonging to the autumn and winter months (Fig. 10). In Marmara, highest daily mean extreme precipitation is shown to occur under the convective types, and followed by Black-Sea effected and cyclonic CTs. During convective activity, we showed a peak during afternoon hours of the day. Main reason of this event can be the diurnal heating and this is further investigated in the next section. For the Black Sea effected EPs, we observe an hourly peak of the precipitation close to noontime and this suggests that when maximum solar radiation reaches the sea surface, significant amount of moisture and heat are transferred by northerly flows to Marmara, generating a considerable amount of precipitation (Baltacı et al. 2015, 2017). During the cyclonic CT, the region takes dense hourly precipitation at the mid-afternoon of the day. In regard to Aegean in winter, cyclones generally release dense precipitation potentials from night to noon times.

## 3.4 Relationship between extreme daily precipitation and surface temperature

From the previous studies, it can be said that the link between precipitation intensity and temperature was explained by Clausius-Clapeyron (C-C) relation. C-C relation presents the moisture-holding capacity of the atmosphere to temperature, hinting a roughly 7% increase in atmospheric moisture storage per degree Celcius. Pall

et al. (2007) found a high agreement between the C-C relation and the changes in the rainfall extremes at midlatitudes. Lenderink and van Meijgaard (2008) found for Netherlands that changes in hourly and daily precipitation intensity generally increased at the 7% °C$^{-1}$ rate anticipated by the C-C at temperatures below 10 °C, but that hourly precipitation exhibited a "super C-C" relation (increase greater than 7% °C$^{-1}$). Later, Lenderink and van Meijgaard (2009) considered that stronger updrafts due to greater latent heat release are the main physical mechanism in the formation of the super C-C relationships. Haerter and Berg (2009) suggested that super C-C scaling may be prevalent in regions that have a relatively balanced co-existence of both convective and large-scale rainfall events.

In this part, to examine C-C relation for the convective EPs, we extracted 10 daily mean temperature records and extreme hourly precipitation records of the extreme precipitation days for the selected south (Bursa) and east (Kocaeli) stations of Marmara in autumn (Table 3). It was shown that hourly extreme precipitation is more linked to daily mean temperature (r=0.69, statistically significant at 95% confidence level) in the south of Marmara under the proper synoptic conditions and daily mean temperature changes from 12.2 to 18.4 °C.

## 4. Summary and Conclusions

In this study, a 10-yr (2006-2015) climatology of EPEs in the west Turkey was developed using hourly precipitation values of 51 and 47 stations in Marmara and Aegean regions, respectively. To define extreme precipitation, we used geographically varying thresholds based upon the 90[th] percentile of 24-h precipitation at each station. The characteristics of the EPEs in each region were analyzed objectively using Lamb Weather Type (LWT) methodology and radar products. Physical mechanisms behind

EPEs were evaluated by considering synoptic-scale composites of NCEP/NCAR Reanalysis daily mean sea level pressure, sea surface temperature, and air temperature at 850-hPa. Salient results of the analysis are as follows:

- While highest EP threshold limits are shown to exist at the seaside stations of western Turkey (above 80 mm), the lowest limits are observed at the semi-arid continental areas of the Aegean and Marmara regions. Seasonal numbers of the EP days showed that Marmara and Aegean areas of Turkey are more influenced from these intense rainfall episodes during autumn and winter months, respectively.

- During autumn, convective, cyclonic and sea-effect originated EPEs represent 21%, 17%, and 15% of total extreme precipitation numbers occurring in the stations of Marmara. If the region has the proper synoptic conditions (HPC over Balkan Peninsula and LPC over eastern Mediterranean) and diurnal heating, convective types of EP mainly occur at the south of Marmara during afternoon and evening times of the day. Daily extreme precipitation amounts are more common in the southwestern parts of Marmara when the cyclone is located over Marmara. Additionally, as a consequence of the interaction between HPC over eastern Europe and LPC over central Anatolia, strong moisture can be transferred by the northeasterly flows and this can result in higher daily precipitation records that was sea-effect originated are shown to develop at the northeast parts of Marmara.

- At Aegean region, 61% of the total EPs occur from the cyclonic activity during winter and torrential rainfall is found to be experienced at the majority of the stations, especially those located in the south. This condition can be explained

by cold air transfer from north that meet with the relatively warm Aegean Sea
and thus, convergence of warm air above the cold air generates cyclogenesis
which results in heavy precipitation.

We conclude by noting that the methods and the results of the current study can serve as
a basis for future research related to EPEs in the western Turkey and elsewhere. The
methods applied to identify EPEs can be adopted for use in other geographical regions
in Turkey.

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

40

45

Table 1. Total extreme precipitation numbers over Marmara considering 51 stations and their percentage frequency distribution according to Black Sea-effect, cyclonic and convective precipitation types for the period 2006-2015. The total numbers of the days causing EPEs are shown in parenthesis.

| | Total extreme precipitation numbers | Black Sea-effect EPEs | | Cyclonic EPEs | Convective EPEs | | | Other CTs |
|---|---|---|---|---|---|---|---|---|
| **Season/CTs** | | N | NE | C | E | SE | S | |
| Winter | 109 | 0 | 1% (1) | 33% (16) | 12% (5) | 0% | 0% | 54% |
| Spring | 43 | 0% | 7% (2) | 33% (9) | 7% (3) | 0% | 9% (1) | 44% |
| Summer | 65 | 8% (2) | 32% (13) | 23% (11) | 2% (1) | 0% | 0% | 35% |
| Autumn | 267 | 0% | 15% (14) | 17% (18) | 21% (11) | 6% | 1% | 50% |

Table 2. Seasonal total extreme precipitation numbers over Aegean considering 46 stations and their percentage frequency distribution according to Aegean Sea-effect, cyclonic and convective precipitation types for the period 2006-2015. The total numbers of the days causing EPEs are shown in parenthesis.

| | Total extreme precipitation numbers | Aegean Sea-effect EPEs | | Cyclonic EPEs | Convective EPEs | | Other CTs |
|---|---|---|---|---|---|---|---|
| **Season/CTs** | | W | SW | C | E | SE | |
| Winter | 200 | 7% (2) | 11% (4) | 61% (35) | 0% | 0% | 21% |
| Spring | 52 | 8% (3) | 17% (3) | 21% (7) | 4% (3) | 0% | 50% |
| Summer | 24 | 8% (1) | 0% | 17% (3) | 4% (1) | 0% | 71% |
| Autumn | 180 | 1% (1) | 18% (9) | 43% (28) | 2% (3) | 2% | 34% |

Table 3. Daily mean temperature and extreme hourly precipitation records and their temporal correlations during the 10 convective extreme precipitation days in the southern (Bursa) and eastern (Kocaeli) stations. The bold value indicates the statistically significant at the 95% confidence level according to Student's t-test.

| Kocaeli | | Bursa | |
|---|---|---|---|
| Precipitation | Temperature | Precipitation | Temperature |
| 27.6 | 14.7 | 6.6 | 12.2 |
| 7.4 | 12.1 | 18.8 | 15.9 |
| 9.0 | 12.3 | 33.8 | 18.4 |
| 7.8 | 11.0 | 20.4 | 16.3 |
| 14.8 | 15.0 | 14.4 | 16.8 |
| 6.8 | 15.7 | 9.8 | 12.9 |
| 11.6 | 12.9 | 7.4 | 15.2 |
| 16.0 | 13.9 | 11.0 | 12.5 |
| 12.6 | 15.6 | 5.0 | 16.2 |
| 11.0 | 12.5 | 7.0 | 12.8 |
| r=0.40 | | **r=0.69** | |

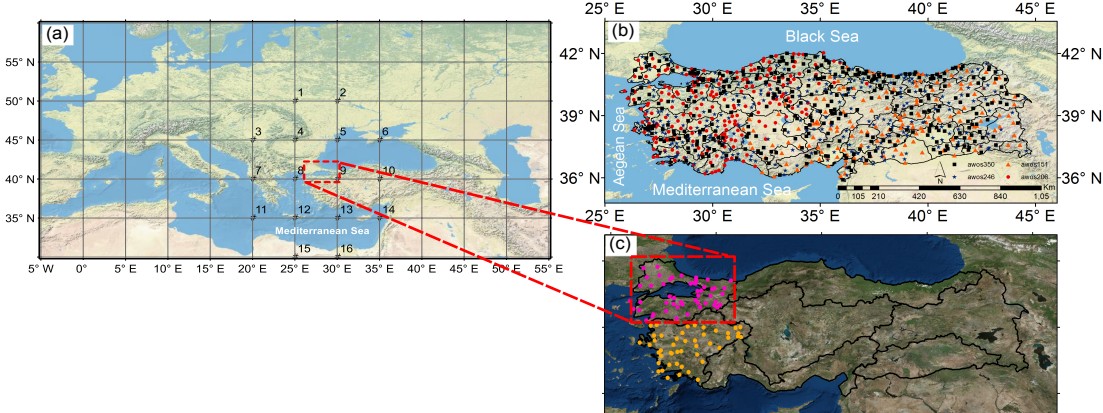

**Figure 1. (a)** The 16 MSLP grid points used in the Lamb Weather Type analysis. The dashed rectangle covers the Marmara Region. **(b)** The distribution of totally 953 automatic weather observing systems (AWOS) over Turkey depending on the four projects (AWOS 206, 151, 246 and 350) and **(c)** the locations of the 51 (pink points) and 47 (light brown points) AWOS stations, at Marmara and Aegean regions. Hourly precipitation data of these 97 stations were provided by the Turkish State Meteorological Service (TSMS) for the period of 2006-2015.

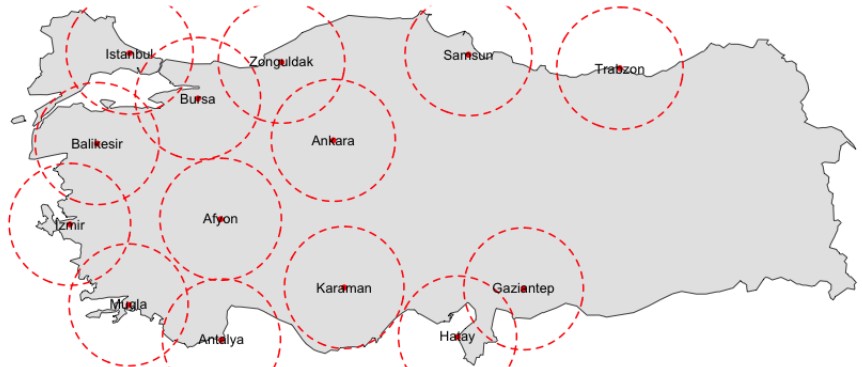

**Figure 2.** The distribution of 14 radar network over Turkey. Precipitation products of six radars (Istanbul, Bursa, Balikesir, Izmir, Mugla, and Afyon), which were taken from TSMS were evaluated manually to describe the characteristics of the precipitation types.

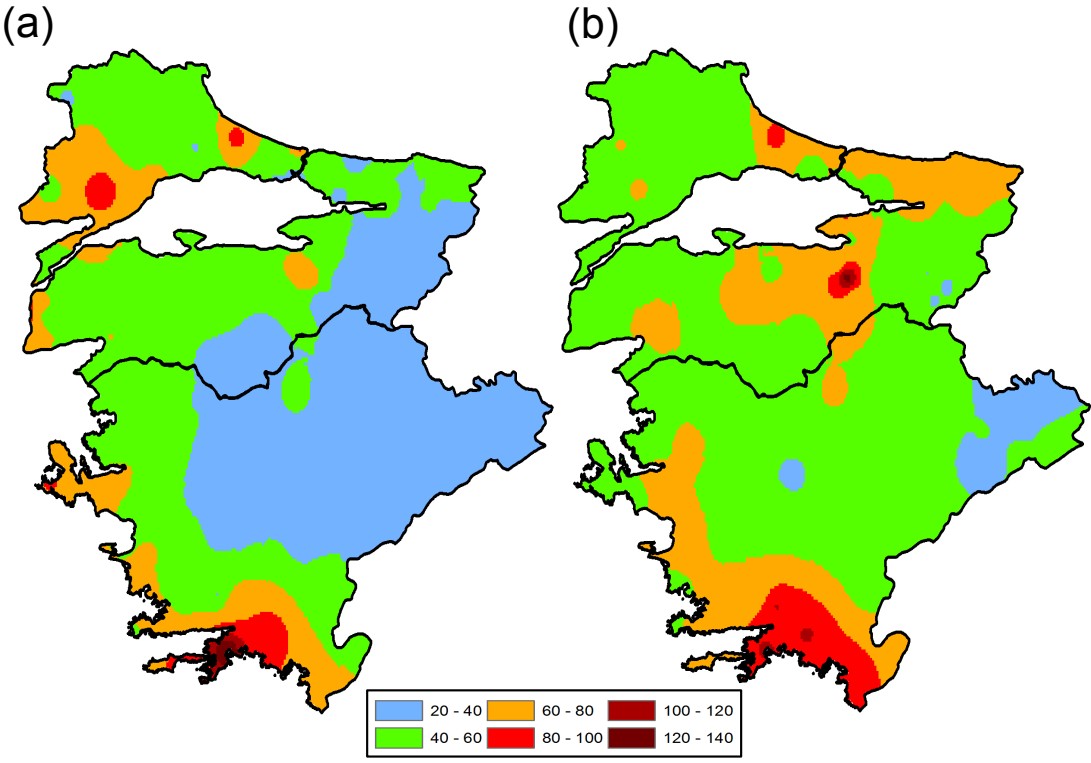

**Figure 3. (a)** Maps show the threshold values (in mm) of the stations during 2006-2015 when precipitation exceeded the 90$^{th}$ percentile generating an EPE. **(b)** the contribution of total EPs of a station to its annual mean precipitation (mm).

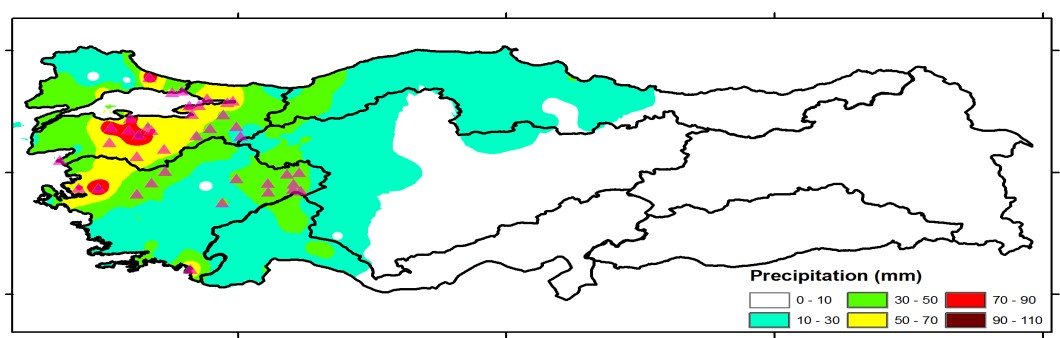

**Figure 4.** Daily total precipitation amounts over Turkey on October 28, 2010 (mm, in shaded) and the stations exceeding their 90[th] percentile threshold (triangle).

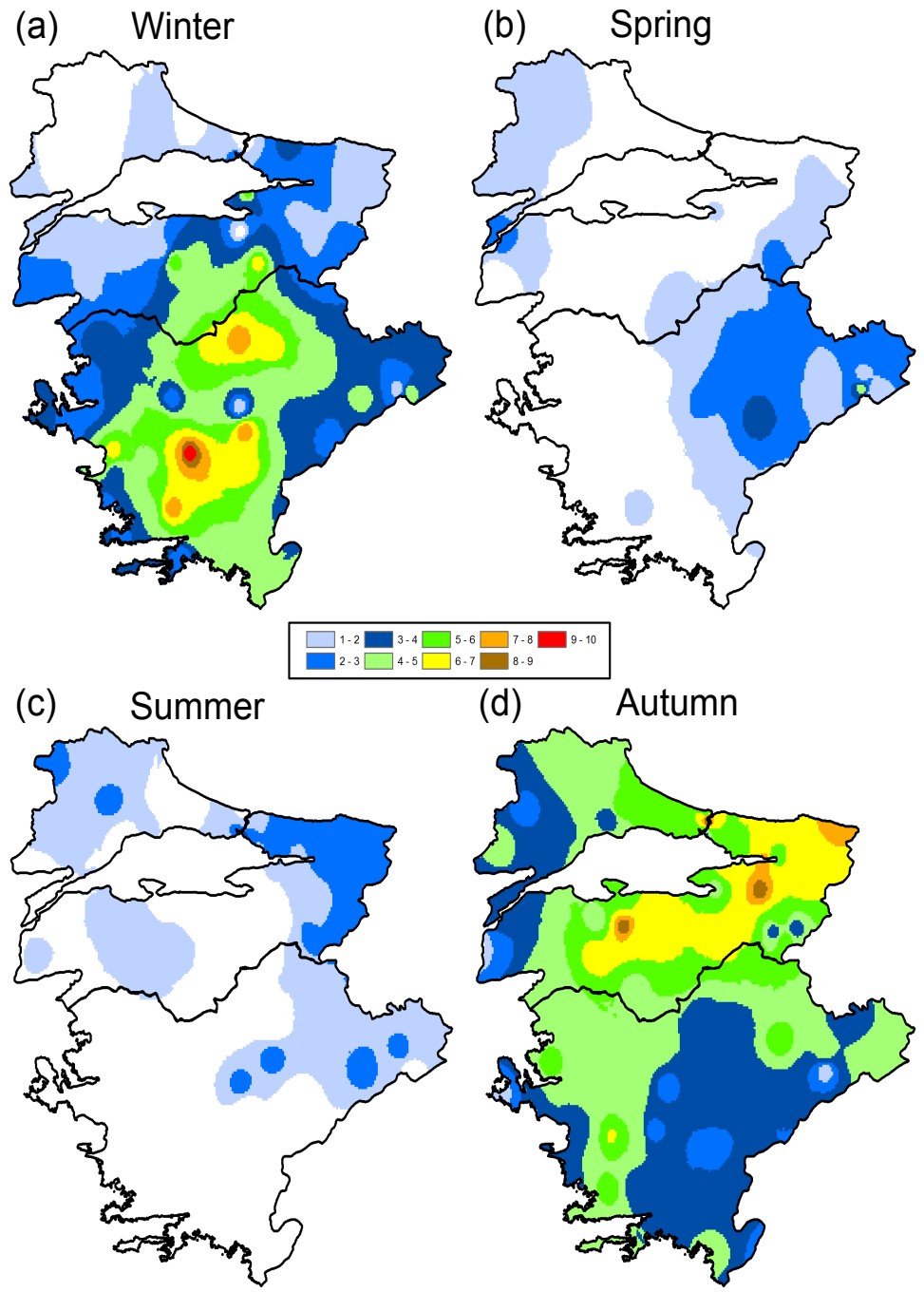

**Figure 5.** Seasonal distribution of the counts of the days for the stations when precipitation exceeded their 90th percentile during an EPE case for **(a)** winter, **(b)** spring, **(c)** summer, and **(d)** autumn seasons.

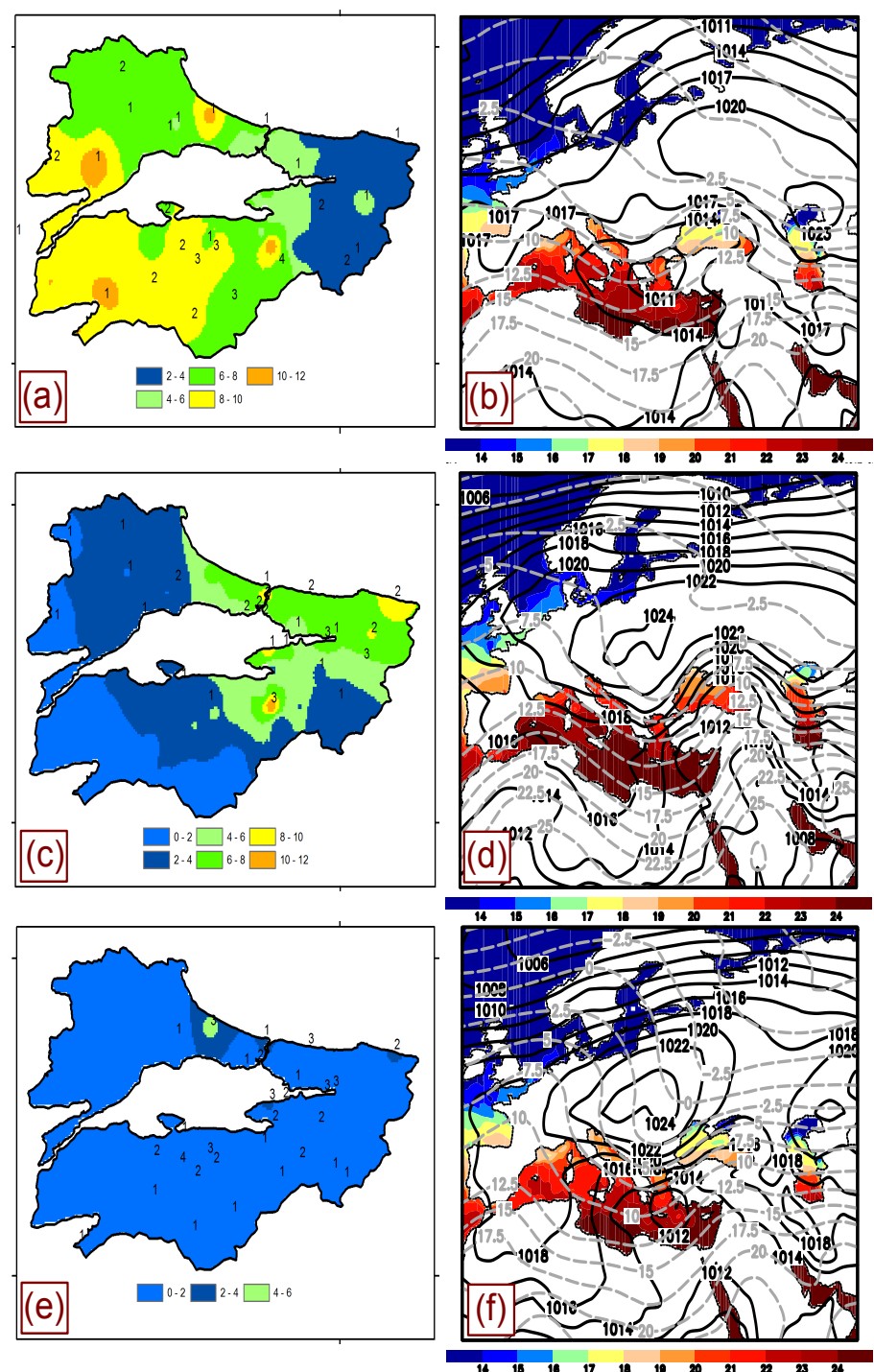

**Figure 6. (a)** Daily mean precipitation values of cyclonic precipitation types (mm, shaded) and the counts of EP days for the stations of Marmara during the autumn of 2006-2015. **(b)** Composites of the daily mean sea level pressure (MSLP, solid lines), sea surface temperature (SST, colored), and air temperature at 850-hPa (dashed lines) for the average of 18 extreme precipitation days over Marmara. **(c)** same as **(a)** but for the sea-effect (NE) precipitation types. **(d)** same as **(b)** but for the 14 extreme precipitation days. **(e)** same as **(a)** but for the convective (E) precipitation types. **(f)** same as **(b)** but for the 11 extreme precipitation days.

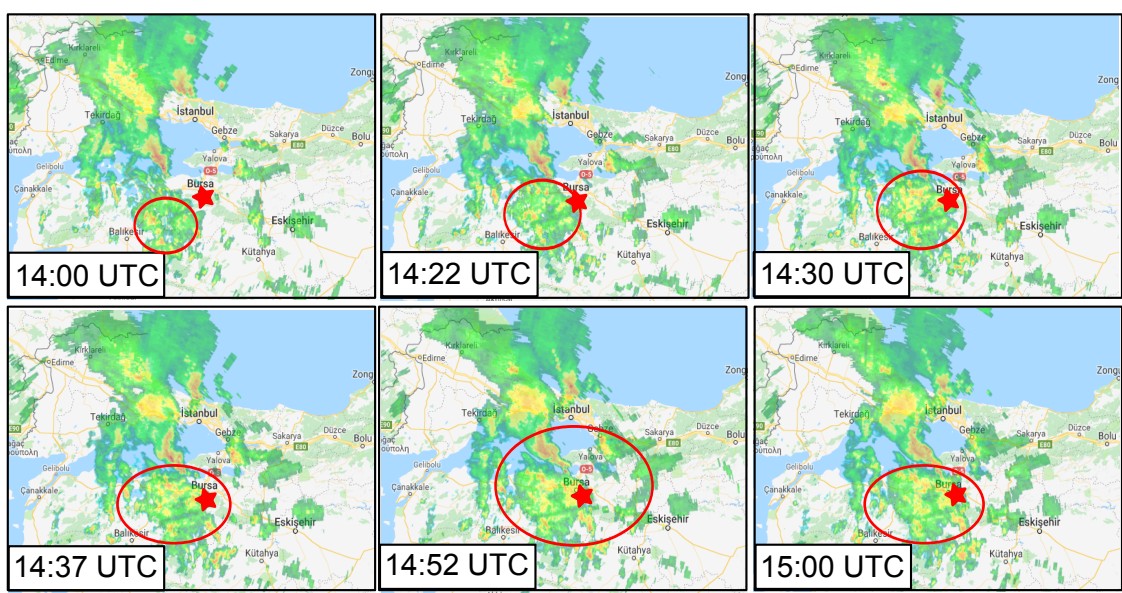

**Figure 7.** Balikesir radar PPI (Plan Position Indicator) image of the Marmara Region on 28 September 2015. Red star marks the Mt. Uludag.

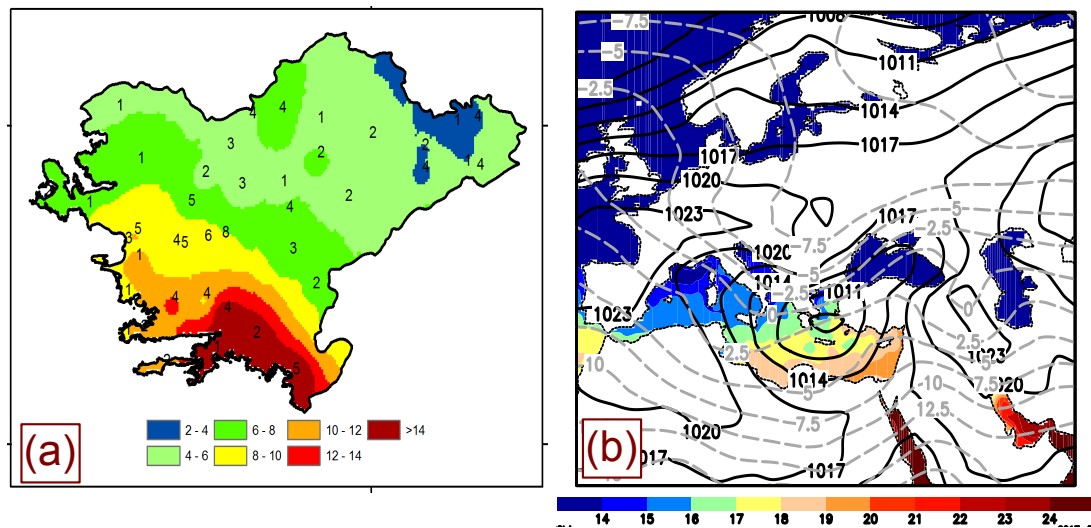

**Figure 8. (a)** Daily mean precipitation values of cyclonic precipitation types (mm, shaded) and the counts of EP days of the Aegean stations for the winter months during 2006-2015. **(b)** Composites of the daily mean sea level pressure (MSLP, solid lines), sea surface temperature (SST, shaded), and air temperature at 850-hPa (dashed lines) for the average of 35 extreme precipitation days over Aegean.

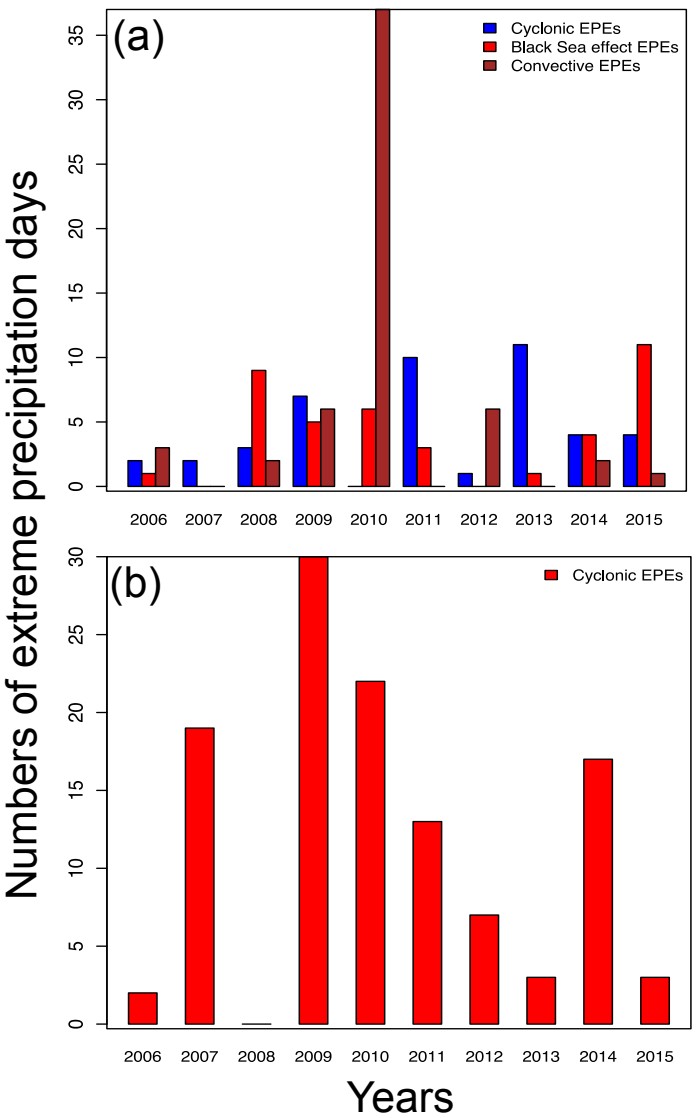

**Figure 9.** Annual distribution of the total counts of EPEs as well as precipitation characteristics for **(a)** Marmara in autumn and **(b)** Aegean in winter months.

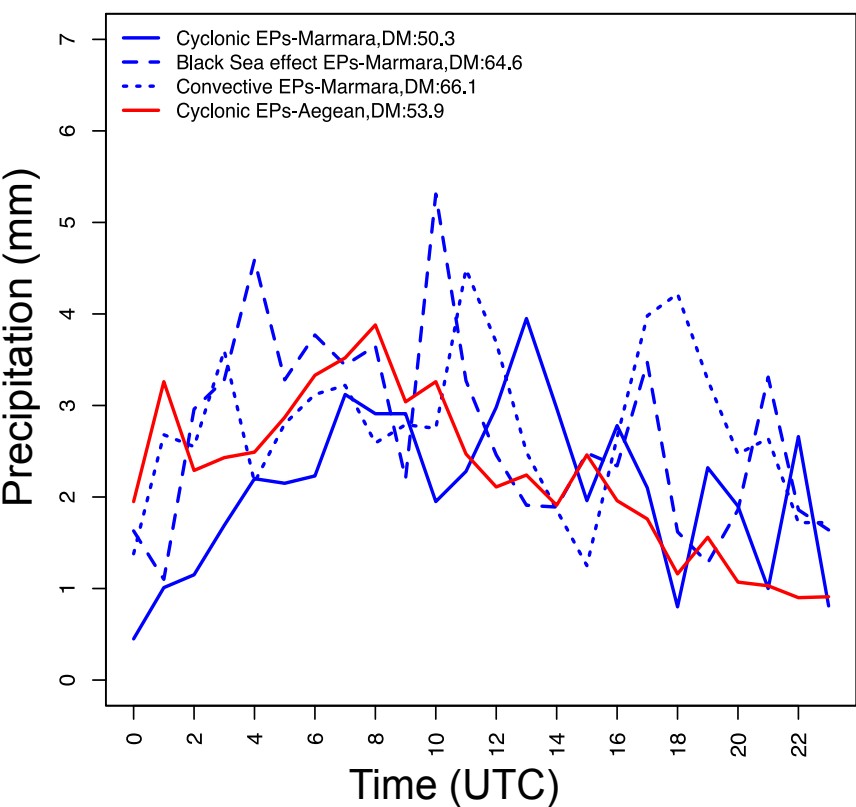

**Figure 10.** Average hourly precipitation amounts (mm) of EP days according to cyclonic, Black-Sea affected and convective types in Marmara for autumn and cyclonic EPs for Aegean in winter. DM indicates the daily mean precipitation amounts (mm) associated with the count of days ended-up with extreme precipitation.