# Peer review of "Discussion started: 20 March 2018 © Author(s) 2018. CC BY 4.0 License."

_Natural Hazards and Earth System Sciences, 2018_

## Short Comment (SC1) · 8 Apr 2018

Author(s) have to taken into account the following comments:

- Title: the second letter shoul initiate with lower case letter; "climatology" instead of "Climatology".

- Keywords are missed, they should be included at the end of the abstract.

- Threshold value of dT,( temperature differences between T850hPa and SST which is equal to13 degrees Celcius) should be compared with other references. Comparisions have to be discussed at this part of the paper.

[Figure]

- The list of acrynoms should be added at the first page of the paper.

- At Figs. 2, 3, 4, 5, 6 and 7; North direction and scale should be added.

- Cyclonic EPS bars should be in the same colour (blue) in two regions at Fig. 8.

---

## Referee Comment (RC1) · Anonymous Referee #1 · 2 Jul 2018

This paper attempts to investigate the precipitation types (using radar outputs and circulation weather types) of extreme precipitation events (EPEs) over western Turkey for the period 2006-2015. This work grouped weather types by origin sea effect, cyclonic effect and convective effect based on Lamb Weather Classification and helping with radar data. The final objective is a descriptive analysis of the EPEs patterns in terms of spatio-temporal and environmental point of view. Although I find the angle really interesting and worthy of research I think that this paper should go through to a major revision process before being published as there are strong issues related to the methodology and some unclear sections that needs major revisions. Overall, this work has potential to develop a good article but is not mature for the publication in an

international journal at this stage. The way in which it is written is hard to read. There are many redundancies in terms of writing style, paper structure and most importantly big sentences with complex meaning. I would recommend to get this paper edited by a professional editor/corrector. My main concerns are Introduction and title: - Talking about climatology for a period of 10 years is not correct. I recommend you another title including the period (2006-2010) without climatology. - You need references or justify better why you are choosing percentile 90th to talk about extreme events. - In the abstract you have identified precipitation types by using radar information and LWT but in the title you are budding between precipitation types and atmospheric conditions. - From line 20 the abstract is confused. I recommend you to rewrite the highlights of the abstract. I suggest extensively to revise the Abstract to better summarize the paper. Some information is missing in the current version. - I suggest to the authors thoroughly review previous studies and summarize their limitations, based on which the specific objective of this paper should be explicitly presented. Without this information, readers cannot evaluate the novelty of the contribution of the paper. You need to add more references in the first paragraph. The lack of flow and structuring of ideas makes it weak. E.g. you do not reference to studies that use radar to characterize precipitation types. Data and methodology: - I recommend you to do a new section/subsection of study area (if you are not from the region it can be confused) highlighting the orography, regions, and some useful and generic information. - It is not clear how you use the radar data. You are talking about radar in the methodology but you don't mention the radar analysis in the results. - Lamb Weather Type is not an objective classification. Jenkinson and Collison (based on LWT) is the objective classification. For this study you have to apply Jenkinson and Collison (and show the equations and the grid points used) or use a new methodology based on ACP/cluster analysis. Using a subjective methodology is not very rigorous. - Why do you use mean daily pressure data? Is it possible that sub-daily changes in the circulation may distort the pattern when averaged? You are working with hourly resolution and convective weather types are difficult to reflect in a "daily mean synoptic scale". - In section 2.4:

there is a lack of examples or studies that use percentiles to establish thresholds of extreme precipitation, is not enough the example of Karl et al., 1996. In this section the methodology is unclear and needs more scientific rigor. Results: - I suggest to create a discussion chapter. You are talking about discussion but there are few references. I recommend you to move the discussion paragraphs and extend this new section (after the results). - When the results are read, the methodology becomes clearer, which is why the major revisions focus on section 2, which will influence changes in section 3. However, this section is rather structured. Convection is not a synoptic process. you have to take it into account and not confuse it, as shown throughout the article. - I suggest a new table summarizing all kind of weather types for all EPEs. All kind of weather types that you comment are part of the sample? - You talk about radar in the methodology but is unclear that use radar data for the final results. You can support with the radar data when you talk about convection. - In section 3.1 I don't understand why you talk about a case of study. You are describing the climatology of EPEs (general results, not specific case). - In the figures 3, 5, 6a-c-e and 7a. Which kind of interpolation do you use? I recommend you kriging interpolation for this kind of area and data.

Please also note the supplement to this comment:
https://www.nat-hazards-earth-syst-sci-discuss.net/nhess-2018-29/nhess-2018-29-RC1-supplement.pdf

---

## Referee Comment (RC2) · Anonymous Referee #2 · 31 Jul 2018

**General comments**

In this study, precipitation characteristics (PCs) causing extreme precipitation events (EPEs) are mainly investigated for the western part of Turkey by using Lamb Weather Type (LWT) approach and Radar outputs. In addition to the usage of high spatial resolution meteorological network, the authors divided the PCs as sea-effect, cyclonic and convective EPEs by using methodologies based on LWT. The scientific point of view of this manuscript is really worth to evaluate and this manuscript should be accepted after some minor revisions given below.

**Specific comments**

[Figure]

- To explain the characteristics of the precipitation events, the usage of the LWT should be explained in more detail for the two different regions of Turkey,

- There are some deficiencies in discussion part of the study. I suggest to the authors focusing on different studies based on precipitation mechanism over Eastern Mediterranean Basin and discuss their results with the others,

- Finally, I recommend to use radar outputs for some specific convective cases by using the movements of the precipitation bands to emphasize the topographical importance of the region.
* * *

---

## Author Comment (AC1) · 14 Aug 2018

We thank the reviewer for her short comments for the article. According to your suggestions, we rearranged the manuscript

-Title: the second letter should initiate with lower case letter; "climatology" instead of "Climatology". We changed title as "Atmospheric conditions during extreme precipitation events in western Turkey for the period 2006-2015".

-Keywords are missed; they should be included at the end of the abstract. It is not necessary using keywords for this journal and we did not add any specific key words.

-Threshold value of dT, (temperature differences between T850hPa and SST which is equal to 13 degrees Celcius) should be compared with other references. Comparisons have to be discussed at this part of the paper.

To indicate the importance of the SST-T850, we added some references explaining convective instability as follows, and later we discussed main findings with the previous study published by Baltaci et al. (2015): "It is shown from the previous studies that the primary factor for the formation and intensity of sea-effect precipitation is known to be the temperature difference between sea surface and the air at 850 hPa level (Holroyd, 1971; Niziol, 1987; Steenburgh et al., 2000). If the SST-T850 difference becomes higher, the chance of precipitation increases due to higher convective instability. At the later study. . ."

-The list of acronyms should be added at the first page of the paper. Done.

-At Figs. 2, 3, 4, 5, 6 and 7; North direction and scale should be added. We added North direction and scale bar for the first figure. To refrain from repetition, we did not use for the later figures.

-Cyclonic EPs bars should be in the same colour (blue) in two regions at Fig. 8. We thank the reviewer for her alert for the figure. However, to compare the different behaviors of EPs between Aegean and Marmara, we showed hourly distribution of EPs with different line colours in Fig. 10. and thus, we do not change the colour bars in Fig. 9.

---

## Author Comment (AC2) · 14 Aug 2018

This paper attempts to investigate the precipitation types (using radar outputs and circulation weather types) of extreme precipitation events (EPEs) over western Turkey for the period 2006-2015. This work grouped weather types by origin sea effect, cyclonic effect and convective effect based on Lamb Weather Classification and helping with radar data. The final objective is a descriptive analysis of the EPEs patterns in terms of spatio-temporal and environmental point of view. Although I find the angle really interesting and worthy of research I think that this paper should go through to a major revision process before being published as there are strong issues related to

the methodology and some unclear sections that needs major revisions. Overall, this work has potential to develop a good article but is not mature for the publication in an international journal at this stage. The way in which it is written is hard to read. There are many redundancies in terms of writing style, paper structure and most importantly big sentences with complex meaning. I would recommend to get this paper edited by a professional editor/corrector. My main concerns are Introduction and title:

-Talking about climatology for a period of 10 years is not correct. I recommend you another title including the period without climatology.

According to your comments, we changed title as: "Atmospheric conditions during extreme precipitation events in western Turkey for the period 2006-2015"

- You need references or justify better why you are choosing percentile 90th to talk about extreme events.

We added some references using 90th percentile and we rearranged related part as follows: "For this reason, similar to the previous studies (Jones, 2000; Zhang et al., 2001; Piccarreta et al., 2013; Krichak et al., 2014), we chosen a methodology…"

- In the abstract you have identified precipitation types by using radar information and LWT but in the title you are budding between precipitation types and atmospheric conditions.

You are right. Therefore, we changed title as "Atmospheric conditions during extreme precipitation events in western Turkey for the period 2006-2015"

-From line 20 the abstract is confused. I recommend you to rewrite the highlights of the abstract. I suggest extensively to revise the Abstract to better summarize the paper. Some information is missing in the current version.

According to your comments we rearranged related parts as follows: "While convective EPEs are seen more common in the southern portions, cyclonic and sea effect originated EPs are mainly affect the southwest and northeastern parts of Marmara.

Among these three precipitation types, convective CTs produce more intense daily precipitation (66.1 mm in average) in the Marmara region due to the interaction between high-pressure center over Balkan Peninsula and low-pressure center over eastern Mediterranean. Based on the hourly observations, convective types of EP show two peak values during afternoon and evening times of the day and are linked to diurnal heating. . ."

-I suggest to the authors thoroughly review previous studies and summarize their limitations, based on which the specific objective of this paper should be explicitly presented. Without this information, readers cannot evaluate the novelty of the contribution of the paper. You need to add more references in the first paragraph. The lack of flow and structuring of ideas makes it weak. E.g. you do not reference to studies that use radar to characterize precipitation types.

According to your suggestions, we developed and added more references to the first part, and later, we emphasized the importance of this paper at the last part.

Data and Methodology

-I recommend you to do a new section/subsection of study area (if you are not from the region it can be confused) highlighting the orography, regions, and some useful and generic information.

Thanks the reviewer for the valuable knowledge. Instead of new subsection, we added properties of Marmara and Aegean to the end of the Section 2.1 as follows: "From these two regions, Marmara has different climatic characteristics in itself. While inland areas have temperate continental climate, milder climate of places on the Black Sea coast resembles more of an oceanic climate, typical to other areas of Turkish Black Sea coast. The coasts in Marmara and Aegean parts have Mediterranean (Med) climate and this region is second smallest Turkish region in size after Southeastern Anatolia. Only south and east parts of the region are more mountainous. In terms of Aegean, this region has a Med climate with mean annual precipitation changing from 450 to 1200

mm yr-1 (Asikoglu and Benzeden 2014). Although climatic behavior of the Aegean is similar to the Med climate, it is shown obvious differences in landscape. Unlike the more parallel mountains found along the Med, Aegean mountains often cut directly into the sea."

-It is not clear how you use the radar data. You are talking about radar in the methodology but you don't mention the radar analysis in the results.

Thanks a lot for your alert. We used radar data in the study and as an example for convective activity, we added new Figure and sentences to the end of the section 3.3.

-Lamb Weather Type is not an objective classification. Jenkinson and Collision (based on LWT) is the objective classification. For this study you have to apply Jenkinson and Collision (and show the equations and grid points used) or use a new methodology based on ACP/Cluster analysis. Using a subjective methodology is not very rigorous.

Thanks for your alert. In this study, we used objective version of LWT and in accordance with your suggestions, we added new figure (Fig. 1a) and we rearranged this part together with explaining the usage of LWT and its equations in the section 2.3.

-Why do you use mean daily pressure data? Is it possible that sub-daily changes in the circulation may distort the pattern when averaged? You are working with hourly resolution and convective weather types are difficult to reflect in a "daily mean synoptic scale".

It is known that daily mean sea level pressure is extracted from the averages of 6-hourly sea level pressure data. In the study, hourly precipitation dataset is firstly converted to the daily totals and then we focused convective types, which are dominated in the particular day.

-In Section 2.4: there is a lack of examples or studies that use percentiles to establish thresholds of extreme precipitation, is not enough the example of Karl et al., 1996. In this section the methodology is unclear and needs more scientific rigor.

You are right. In this section, we added extra references to define fixed thresholds. Then we added references indicating percentile based precipitation thresholds and we explained why we chosen 10% of largest daily precipitation amounts due to the large differences in topography and irregularity in precipitation distribution.

Results:

-I suggest to create a discussion chapter. You are talking about discussion but there are few references. I recommend you to move the discussion paragraphs and extend this new section (after the results)

In this study, we developed results section by adding new references and we discussed our results with the precious ones.

-When the results are read, the methodology becomes clearer, which is why the major revisions focus on section 2, which will influence changes in section 3. However, this section is rather structured. Convection is not a synoptic process. You have to take it into account and not confuse it, as shown throughout the article.

We rearranged the results and methodology section in accordance with your suggestions. Also, we tried to explain convective instability under some particular synoptic processes by using radar outputs and equations

-I suggest a new table summarizing all kind of weather types for all EPEs. All kind of weather types that you comment are part of the sample?

Thanks the reviewer. We emphasized the most important CTs that cause cyclonic, sea effect and convective precipitation in the manuscript. Therefore, to not cause confusion, we did not add all CTs for all EPEs.

-You talk about radar in the methodology but is unclear that use radar data for the final results. You can support with the radar data when you talk about convection.

You are right. We supported convectivity with radar by given an example related to 28

September 2015.

-In section 3.1 I don't understand why you talk about a case of study. You are describing the climatology of EPEs (general results, not specific case).

We given this section, because when we defined threshold values of extreme precipitation for each station, it can be shown that although some stations take less daily precipitation when compared with others, their daily precipitation values exceed their threshold limits (Triangles in Fig. 4).

-In the figures 3, 5, 6a-c-e and 7a. Which kind of interpolation do you use? I recommend you kriging interpolation for this kind of area and data. We used krigging interpolation for this figures.

Please also note the supplement to this comment:
https://www.nat-hazards-earth-syst-sci-discuss.net/nhess-2018-29/nhess-2018-29-AC2-supplement.pdf

[revised manuscript text omitted]

**mac baltaci 14/8/2018 20:54**

**mac baltaci 12/8/2018 17:34**

25° E

(a)

42° N

39° N

36° N

25° E

(b)

**Figure 1. (a)** The dist
(AWOS) over Turkey
and **(b)** the locations
at Marmara and Aege
provided by the Turki
2015.

[Figure]

**Figure 2.** The distribution of 14 radar network over Turkey. Precipitation products of six radars (Istanbul, Bursa, Balikesir, Izmir, Mugla, and Afyon), which were taken from TSMS were evaluated manually to describe the characteristics of the precipitation types.

[Figure]

**Figure 3. (a)** Maps show the threshold values (in mm) of the stations during 2006-2015 when precipitation exceeded the 90[th] percentile generating an EPE. **(b)** the contribution of total EPs of a station to its annual mean precipitation (mm).

[Figure]

**Figure 4.** Daily total precipitation amounts over Turkey on October 28, 2010 (mm, in shaded) and the stations exceeding their 90th percentile threshold (triangle).

[Figure]

**Figure 5.** Seasonal distribution of the counts of the days for the stations when precipitation exceeded their 90th percentile during an EPE case for **(a)** winter, **(b)** spring, **(c)** summer, and **(d)** autumn seasons.

[Figure]

**Figure 6. (a)** Daily mean precipitation values of cyclonic precipitation types (mm, shaded) and the counts of EP days for the stations of Marmara during the autumn of 2006-2015. **(b)** Composites of the daily mean sea level pressure (MSLP, solid lines), sea surface temperature (SST, colored), and air temperature at 850-hPa (dashed lines) for the average of 18 extreme precipitation days over Marmara. **(c)** same as **(a)** but for the sea-effect (NE) precipitation types. **(d)** same as **(b)** but for the 14 extreme precipitation days. **(e)** same as **(a)** but for the convective (E) precipitation types. **(f)** same as **(b)** but for the 11 extreme precipitation days.

[Figure]

**Figure 7.** Balikesir radar PPI (Plan Position Indicator) image of the Marmara Region on 28 September 2015. Red star marks the Mt. Uludag.

[Figure]

[Figure]

[Figure]

**Figure 8. (a)** Daily mean precipitation values of cyclonic precipitation types (mm, shaded) and the counts of EP days of the Aegean stations for the winter months during 2006-2015. **(b)** Composites of the daily mean sea level pressure (MSLP, solid lines), sea surface temperature (SST, shaded), and air temperature at 850-hPa (dashed lines) for the average of 35 extreme precipitation days over Aegean.

mac baltaci 13/8/2018 14:41

**Figure 7. (a)** Daily mean pr and the counts of EP days **(b)** Composites of the daily temperature (SST, shaded) of 35 extreme precipitation

[Figure]

**Figure 9.** Annual distribution of the total counts of EPEs as well as precipitation characteristics for **(a)** Marmara in autumn and **(b)** Aegean in winter months.

[Figure]

mac baltaci 13/8/2018 14:41

**Figure 8.** Annual distributio characteristics for **(a)** Marm

[Figure]

**Figure 10.** Average hourly precipitation amounts (mm) of EP days according to cyclonic, Black-Sea affected and convective types in Marmara for autumn and cyclonic EPs for Aegean in winter. DM indicates the daily mean precipitation amounts (mm) associated with the count of days ended-up with extreme precipitation.

[Figure]

**Figure 9.** Average hourl cyclonic, Black-Sea affe cyclonic EPs for Aegear amounts (mm) associate precipitation.

mac baltaci 13/8/2018 14:41

---

## Author Comment (AC3) · 14 Aug 2018

In this study, precipitation characteristics (PCs) causing extreme precipitation events (EPEs) are mainly investigated for the western part of Turkey by using Lamb Weather Type (LWT) approach and Radar outputs. In addition to the usage of high spatial resolution meteorological network, the authors divided the PCs as sea-effect, cyclonic and convective EPEs by using methodologies based on LWT. The scientific point of view of this manuscript is really worth to evaluate and this manuscript should be accepted after some minor revisions given below.

-To explain the characteristics of the precipitation events, the usage of the LWT should

[Figure]

be explained in more detail for the two different regions of Turkey

Thanks for your alert. In this study, we used objective version of LWT and in accordance with your suggestions, we added new figure (Fig. 1a) and we rearranged this part together with explaining the usage of LWT and its equations in the section 2.3.

-There are some deficiencies in discussion part of the study. I suggest to the authors focusing on different studies based on precipitation mechanism over Eastern Mediterranean Basin and discuss their results with the others.

According to yours and the other reviewers suggestions, we developed results section by adding new references and we discussed our results with the precious ones.

-Finally, I recommend to use radar outputs for some specific convective cases by using the movements of the precipitation bands to emphasize the topographical importance of the region.

You are right. We supported convectivity with radar by given an example related to 28 September 2015.

Please also note the supplement to this comment:
https://www.nat-hazards-earth-syst-sci-discuss.net/nhess-2018-29/nhess-2018-29-AC3-supplement.pdf

———————————————————

[revised manuscript text omitted]

**mac baltaci 14/8/2018 20:54**

**mac baltaci 12/8/2018 17:34**

25° E

(a)

42° N

39° N

36° N

25° E

(b)

**Figure 1. (a)** The dist
(AWOS) over Turkey
and **(b)** the locations
at Marmara and Aege
provided by the Turki
2015.

[Figure]

**Figure 2.** The distribution of 14 radar network over Turkey. Precipitation products of six radars (Istanbul, Bursa, Balikesir, Izmir, Mugla, and Afyon), which were taken from TSMS were evaluated manually to describe the characteristics of the precipitation types.

[Figure]

**Figure 3. (a)** Maps show the threshold values (in mm) of the stations during 2006-2015 when precipitation exceeded the 90[th] percentile generating an EPE. **(b)** the contribution of total EPs of a station to its annual mean precipitation (mm).

[Figure]

**Figure 4.** Daily total precipitation amounts over Turkey on October 28, 2010 (mm, in shaded) and the stations exceeding their 90th percentile threshold (triangle).

[Figure]

**Figure 5.** Seasonal distribution of the counts of the days for the stations when precipitation exceeded their 90th percentile during an EPE case for **(a)** winter, **(b)** spring, **(c)** summer, and **(d)** autumn seasons.

[Figure]

**Figure 6. (a)** Daily mean precipitation values of cyclonic precipitation types (mm, shaded) and the counts of EP days for the stations of Marmara during the autumn of 2006-2015. **(b)** Composites of the daily mean sea level pressure (MSLP, solid lines), sea surface temperature (SST, colored), and air temperature at 850-hPa (dashed lines) for the average of 18 extreme precipitation days over Marmara. **(c)** same as **(a)** but for the sea-effect (NE) precipitation types. **(d)** same as **(b)** but for the 14 extreme precipitation days. **(e)** same as **(a)** but for the convective (E) precipitation types. **(f)** same as **(b)** but for the 11 extreme precipitation days.

[Figure]

**Figure 7.** Balikesir radar PPI (Plan Position Indicator) image of the Marmara Region on 28 September 2015. Red star marks the Mt. Uludag.

[Figure]

[Figure]

[Figure]

**Figure 8. (a)** Daily mean precipitation values of cyclonic precipitation types (mm, shaded) and the counts of EP days of the Aegean stations for the winter months during 2006-2015. **(b)** Composites of the daily mean sea level pressure (MSLP, solid lines), sea surface temperature (SST, shaded), and air temperature at 850-hPa (dashed lines) for the average of 35 extreme precipitation days over Aegean.

mac baltaci 13/8/2018 14:41

**Figure 7. (a)** Daily mean pr and the counts of EP days **(b)** Composites of the daily temperature (SST, shaded) of 35 extreme precipitation

[Figure]

**Figure 9.** Annual distribution of the total counts of EPEs as well as precipitation characteristics for **(a)** Marmara in autumn and **(b)** Aegean in winter months.

[Figure]

mac baltaci 13/8/2018 14:41

**Figure 8.** Annual distributio characteristics for **(a)** Marm

[Figure]

**Figure 10.** Average hourly precipitation amounts (mm) of EP days according to cyclonic, Black-Sea affected and convective types in Marmara for autumn and cyclonic EPs for Aegean in winter. DM indicates the daily mean precipitation amounts (mm) associated with the count of days ended-up with extreme precipitation.

[Figure]

**Figure 9.** Average hourl cyclonic, Black-Sea affe cyclonic EPs for Aegear amounts (mm) associate precipitation.

mac baltaci 13/8/2018 14:41

---

## Author Response (AR1)

First of all, thank the experts for carefully reviewing our paper, putting forward pertinent comments and detailed suggestions. We analyzed and discussed the problems raised by experts. We have considered them in the revised manuscript and we are re-submitting for your consideration. We believe that these amendments should facilitate the reading of the paper and also clarify some issues which were defined by the reviewers.

Below you will find the responses to the reviewers and detailed explanations about how the suggestions and criticisms raised by the reviewers were taken into account in the revised manuscript.

**Answers to short comments by Zafer Aslan:**

Title: the second letter should initiate with lower case letter; "climatology" instead of "Climatology".
*Response: We changed title as "Atmospheric conditions of extreme precipitation events in western Turkey for the period 2006-2015".*

Keywords are missed; they should be included at the end of the abstract.
*Response: It is not necessary using keywords for this journal and we did not add any specific key words.*

Threshold value of dT, (temperature differences between T850hPa and SST which is equal to 13 degrees Celcius) should be compared with other references. Comparisons have to be discussed at this part of the paper.
*Response: To indicate the importance of the SST-T850, we added some references explaining convective instability as follows, and later we discussed main findings with the previous study published by Baltaci et al. (2015):*
*"It is shown from the previous studies that the primary factor for the formation and intensity of sea-effect precipitation is known to be the temperature difference between sea surface and the air at 850 hPa level (Holroyd, 1971; Niziol, 1987; Steenburgh et al., 2000). If the SST-T850 difference becomes higher, the chance of precipitation increases due to higher convective instability. At the later study…"*

The list of acronyms should be added at the first page of the paper.
*Response: Done.*

At Figs. 2, 3, 4, 5, 6 and 7; North direction and scale should be added.
*Response: We added North direction and scale bar for the first figure. To refrain from repetition, we did not use for the later figures.*

Cyclonic EPs bars should be in the same colour (blue) in two regions at Fig. 8.

*Response: We thank the reviewer for her alert for the figure. However, to compare the different behaviors of EPs between Aegean and Marmara, we showed hourly distribution of EPs with different line colours in Fig. 9. and thus, we do not change the colour bars in Fig. 8.*

**Anonymous Referee #1**

Talking about climatology for a period of 10 years is not correct. I recommend you another title including the period without climatology.
*Response: According to your comments, we changed title as: "Atmospheric conditions during extreme precipitation events in western Turkey for the period 2006-2015"*

You need references or justify better why you are choosing percentile 90[th] to talk about extreme events.
*Response: We added some references using 90[th] percentile and we rearranged related part (P9, Line 10-11) as follows:*
*"For this reason, similar to the previous studies (e.g. Jones, 2000; Zhang et al., 2001; Piccarreta et al., 2013; Krichak et al., 2014), we chosen a methodology…"*

In the abstract you have identified precipitation types by using radar information and LWT but in the title you are budding between precipitation types and atmospheric conditions.
*Response: Thanks. We changed title as "Atmospheric conditions of extreme precipitation events in western Turkey for the period 2006-2015"*

From line 20 the abstract is confused. I recommend you to rewrite the highlights of the abstract. I suggest extensively to revise the Abstract to better summarize the paper. Some information is missing in the current version.
*Response: According to your comments we rearranged related part from line 20 as follows:*
*"… 15% of total autumn EPEs show convective, cyclonic, and sea-effect precipitation characteristics, respectively. While convective EPEs are seen more common in the southern portions, cyclonic and sea effect originated EPEs mainly affect the southwest and northeastern parts of Marmara. Among these three precipitation types, convective mechanisms generally produce more intense daily precipitation (66.1 mm in average) in the Marmara region under the proper synoptic conditions (high-pressure center over Balkan Peninsula and low-pressure center over eastern Mediterranean). Based on the hourly observations, convective types of extreme precipitations (EPs) show two peak values during afternoon and evening times of the day and are linked to diurnal heating. In terms of Aegean region, cyclonic originated EPs, which include 65% of the total winter EPEs, are more common in the whole territory and reach to its peak value during the first hours of the day."*

I suggest to the authors thoroughly review previous studies and summarize their limitations, based on which the specific objective of this paper should be explicitly presented. Without this information, readers cannot evaluate the novelty of the

contribution of the paper. You need to add more references in the first paragraph. The lack of flow and structuring of ideas makes it weak. E.g. you do not reference to studies that use radar to characterize precipitation types.
*Response: Thanks; we developed the first paragraph by also adding more references. Also we gave reference based on radar usage in defining precipitation types. At the latest part of the introduction, we emphasized the importance of this paper at the last part.*

**Data and Methodology**

I recommend you to do a new section/subsection of study area (if you are not from the region it can be confused) highlighting the orography, regions, and some useful and generic information.
*Response: Thanks. We added a new subsection as "2.1.1 Climatic characteristics of Marmara and Aegean regions" under 2.1 Precipitation dataset as follows:*
*"**2.1.1 Climatic characteristics of Marmara and Aegean regions***
*The Marmara Region is located in the northwest of Turkey, between latitudes 29°N and 32°N and longitudes 38°E and 42°E and covers an area of 67000 $km^2$. Marmara has different climatic characteristics in itself. While inland areas have temperate continental climate, milder climate of places on the Black Sea coast resembles more of an oceanic climate, typical to other areas of Turkish Black Sea coast. The coasts in Marmara and Aegean parts have Mediterranean (Med) climate and this region is second smallest Turkish region in size after Southeastern Anatolia. Only south and east parts of the region are more mountainous.*
*In terms of Aegean, this region has a Med climate with mean annual precipitation changing from 450 to 1200 mm $yr^{-1}$ (Asikoglu and Benzeden 2014). Although climatic behavior of the Aegean is similar to the Med climate, it is shown obvious differences in landscape. Unlike the more parallel mountains found along the Med, Aegean mountains often cut directly into the sea."*

It is not clear how you use the radar data. You are talking about radar in the methodology but you don't mention the radar analysis in the results.
*Response: Thanks a lot for your alert. Now, we mentioned of using radar data in the study as an example for convective activity, and we added new Figure and sentences to the end of the section 3.3 as follows.*
*"As an example, convective activity in southern Marmara started afternoon times on 28 September 2015 (Fig. 7). Due to the movements of the single cells to the easterly directions their spatial area expanded. When the cells met with orographic barrier over Bursa (Mt. Uludag, red star in Fig. 7) quasi-stationary conditions developed the convective instability. As a result, extreme precipitation amounts were recorded in the western part of the mountainside in a very short time."*

Lamb Weather Type is not an objective classification. Jenkinson and Collision (based on LWT) is the objective classification. For this study you have to apply Jenkinson and Collision (and show the equations and grid points used) or use a new methodology based on ACP/Cluster analysis. Using a subjective methodology is not very rigorous.
*Response: Thanks. In this study, we used objective version of LWT and in accordance with your suggestions, we added a new figure (Fig. 1a), which explains*

*grid structure and points of LWT approach, and we rearranged this part together with explaining the usage of LWT and its equations in the section 2.3.*

Why do you use mean daily pressure data? Is it possible that sub-daily changes in the circulation may distort the pattern when averaged? You are working with hourly resolution and convective weather types are difficult to reflect in a "daily mean synoptic scale".

*Response: It is known that daily mean sea level pressure (DMSLP) is extracted from the averages of 6-hourly sea level pressure data. In the study, hourly precipitation dataset is firstly converted to the daily totals and then we focused convective types, which are dominated in the particular day by also considering DMSLP.*

In Section 2.4: there is a lack of examples or studies that use percentiles to establish thresholds of extreme precipitation, is not enough the example of Karl et al., 1996. In this section the methodology is unclear and needs more scientific rigor.

*Response: You are right. In this section, we added extra references to define fixed thresholds like this: "(e.g. Brooks and Stensrud 2000; Ralph and Dettinger 2012; Hitchens et al. 2012, 2013)."*
*and we explained why we chosen 10% of largest daily precipitation amounts like this: "For our country, fixed and other threshold limits was not suitable because of the large topographical difference and irregularity of the precipitation distribution." Then we added references indicating percentile based precipitation thresholds as follows: "For this reason, similar to the previous studies (e.g. Jones, 2000; Zhang et al., 2001; Piccarreta et al., 2013; Krichak et al., 2014), we chosen a methodology that defines the threshold levels of each station according to its own precipitation characteristics." In the Section 2.4.*

**Results:**

I suggest to create a discussion chapter. You are talking about discussion but there are few references. I recommend you to move the discussion paragraphs and extend this new section (after the results)
*Response: We developed results section by adding new references and we discussed our results with the precious ones in the sections 3.3 and 3.4.*

When the results are read, the methodology becomes clearer, which is why the major revisions focus on section 2, which will influence changes in section 3. However, this section is rather structured. Convection is not a synoptic process. You have to take it into account and not confuse it, as shown throughout the article.
*Response: We rearranged the results and methodology section in accordance with your suggestions. Also, we tried to explain convective instability under some particular synoptic processes by using radar outputs and equations*

I suggest a new table summarizing all kind of weather types for all EPEs. All kind of weather types that you comment are part of the sample?
*Response: Thanks. We emphasized the most important CTs, which explain 50% and 79% of all fall and winter EPEs in Marmara and Aegean regions, respectively, causing cyclonic, sea effect and convective precipitation types. Therefore, to not cause confusion, we did not add all CTs for all EPEs.*

You talk about radar in the methodology but is unclear that use radar data for the final results. You can support with the radar data when you talk about convection.
*Response: You are right. We supported convectivity with radar by given an example related to 28 September 2015 in the last part of the section 3.3.*

In section 3.1 I don't understand why you talk about a case of study. You are describing the climatology of EPEs (general results, not specific case).
*Response: To emphasize the different threshold limits for each station, we give this section; it can be shown from Fig. 4 that although some stations take less daily precipitation when compared with others, their daily precipitation values exceed their threshold limits.*

In the figures 3, 5, 6a-c-e and 7a. Which kind of interpolation do you use? I recommend you kriging interpolation for this kind of area and data.
*Response: We used krigging interpolation for this figures.*

**Anonymous Referee #2**

-To explain the characteristics of the precipitation events, the usage of the LWT should be explained in more detail for the two different regions of Turkey
*Response: Thanks. In this study, we used objective version of LWT and in accordance with your suggestions, we added a new figure (Fig. 1a), which explains grid structure and points of LWT approach, and we rearranged this part together with explaining the usage of LWT and its equations in the section 2.3.*

-There are some deficiencies in discussion part of the study. I suggest to the authors focusing on different studies based on precipitation mechanism over Eastern Mediterranean Basin and discuss their results with the others.
*Response: According to yours and the other reviewers suggestions, we developed results section by adding new references and we discussed our results with the precious ones.*

Finally, I recommend to use radar outputs for some specific convective cases by using the movements of the precipitation bands to emphasize the topographical importance of the region.
*Response: You are right. We supported convectivity with radar by given an example related to 28 September 2015.*

[revised manuscript text omitted]

mac baltaci 13/8/2018 14:41

[Figure]

**Figure 9.** Average hourl
cyclonic, Black-Sea affe
cyclonic EPs for Aegean
amounts (mm) associate
precipitation.

---

## Referee Report (RR1)

**General comments:**

The article has improved taking into account the comments of all the reviewers. In this case, I propose minor changes so that the article can be accepted.

There continue to be too long and confused sentences. An English native revision would be a solution to correct the entire article.

The Lamb Weather Type classification that used is based on Jenkinson and Collison methodology. You need to clarify it.

In the section 2.4. You identified convective EPEs with a terrestrial origin. But, all of terrestrial EPEs are convective? How do you know. Please explain better the convective events and why you use radar. Convective events are not an events reflected in a synoptic scale. Maybe you have to characterize with another technique like radar, intensity of precipitation…

You only use radar for some convective study cases. But you giving importance the radar in the abstract. Please fix this conflict.

In the methodology you do not explain how analyze the physical mechanisms (850hPa-SST). Explain better how do you use these parameters to describe the instability.

Section 3.5. It is very interesting the result in 2010. You have to explain better the reason of this anomaly. Why was a wet year? Add references in the NAO pattern explanation. Do you read or know any studies relating NAO and 2010 in Turkey?

Why does it rain more in Marmara in autumn and in Aegean in winter? This result of the tables is interesting, in two places of the same territory with this difference. It would be interesting to explain the reason.

Section 4 - Conclusions. Expand the section, summarize the methodology (physical mechanisms) and add a paragraph explaining the importance of the article and its usefulness.

Once the changes are made, try to improve the abstract, it is still a bit confusing.

**Specific Comments:**

Line 21 page 5, delate climatological and use another word.

Line 3 page 6, add another reference, i.e. Zhang and Yang 2004.

Line 4 page 7, delate NWP. You only use numerical weather prediction one time.

Line 10 page 7, Lamb Weather Type Classification, delate technique.

In your paper you use *autumn* but in the figures you use *fall*. Please use one word (in UK you use autumn and in USA fall).

Section 3.1. Delate Climatology, use another title.

Line 16 page 11. Add: As an example, on October…

Section 3.2. The parenthesis "(addition of…)" are confused, add the EPEs after each number.

In sections 3.3 and 3.4. Make it clear that you are analyzing the main EPEs in section 3.2. And for this reason, you only analyze one EPE in section 3.4.

Lines 2 to 8, page 17. It seems a discussion, you can add some references to support your affirmations.

---

## Author Response (AR2)

**General Comments:**

The article has improved taking into account the comments of all the reviewers. In this case, I propose minor changes so that the article can be accepted.
*Response: First of all, thank the reviewer for carefully reviewing our paper, putting forward pertinent comments and detailed suggestions. We rearranged the minor revisions raised by you. We have considered them in the revised manuscript and we are re-submitting for your consideration. We believe that these amendments should facilitate the reading of the paper and also clarify some issues. Below you will find the responses and detailed explanations about how the suggestions and criticisms were taken into account in the revised manuscript.*

There continue to be too long and confused sentences. An English native revision would be a solution to correct the entire article.
*Response: Done.*

The Lamb Weather Type classification that used is based on Jenkinson and Collison methodology. You need to clarify it.
*Response: Thanks. We changed the related sentence as follows:*
*"The subjective version of the Lamb's work (Lamb 1972) was firstly developed as an objective version by Jenkinson and Collison (1977) and refined by Jones et al. (1993) to indicate the circulation types (CTs) influencing British Isles."*

In the section 2.4. You identified convective EPEs with a terrestrial origin. But, all of terrestrial EPEs are convective? How do you know. Please explain better the convective events and why you use radar. Convective events are not an events reflected in a synoptic scale. Maybe you have to characterize with another technique like radar, intensity of precipitation...
*Response: We mainly focused on convective EPEs, which coming from the terrestrial areas, under some specific CTs. To detect and monitor of these convective EPEs, we also used radar data. As a result, under E, SE, and S CTs for Marmara and E and SE types for Aegean were thought to be proper atmospheric CTs to have convective EPEs, which also supported by radar data.*

You only use radar for some convective study cases. But you giving importance the radar in the abstract. Please fix this conflict.
*Response: Cyclonic and sea-effect precipitation types generally are seen in a large area when compared with convective EPEs. Also, convective EPEs generally produce more precipitation in short places (Fig.7) and especially depend on the land properties such as diurnal heating. We used radar data to detect all EPEs together with LWT, however, to indicate the importance of the convective EPEs, we also gave an extra Figure 7 and explanation to the abstract of the manuscript.*

In the methodology you do not explain how analyze the physical mechanisms (850hPa- SST). Explain better how do you use these parameters to describe the instability.

*Response: We added some references and explanations to the manuscript as follows: "Millan et al. (1995) argued that enhanced evaporation resulting from temperature differences between European continental air and the relative warm Mediterranean Sea in fall can become a key factor in determining the onset of precipitation. Pastor et al. (2015) have shown that regions of high heat/moisture air-sea exchange over the Mediterranean Basin are prone to enhancing convection, leading to torrential rain."*

Section 3.5. It is very interesting the result in 2010. You have to explain better the reason of this anomaly. Why was a wet year? Add references in the NAO pattern explanation. Do you read or know any studies relating NAO and 2010 in Turkey?

*Response: According to your suggestions, we added an extra sentence like this:*
*As previously indicated by Türkeş and Erlat (2003) that negative relationships were found between NAO indices and Turkish precipitation series. Statistically significant changes in the precipitation amounts during the extreme NAO phases are more apparent in the west and mid Turkey.*

Why does it rain more in Marmara in autumn and in Aegean in winter? This result of the tables is interesting, in two places of the same territory with this difference. It would be interesting to explain the reason.

*Response: The main reason of this mechanism is shown as the influence of Black Sea-effect precipitation in Marmara during fall and cyclonic EPs in Aegean during winter and atmospheric conditions that trigger these mechanisms were explained in the related sections of the paper.*

Section 4 - Conclusions. Expand the section, summarize the methodology (physical mechanisms) and add a paragraph explaining the importance of the article and its usefulness.

*Response: We added "We conclude by noting that the methods and the results of the current study can serve as a basis for future research related to EPEs in the western Turkey and elsewhere. The methods applied to identify EPEs can be adopted for use in other geographical regions in Turkey." sentence to the end of the Summary and Conclusions.*

Once the changes are made, try to improve the abstract, it is still a bit confusing.
*Response: Not changed.*

**Specific Comments:**

Line 21 page 5, delate climatological and use another word.
*Response: We changed "climatological perspectives" to "atmospheric conditions".*

Line 3 page 6, add another reference, i.e. Zhang and Yang 2004.
*Response: It was added to the related part.*

Line 4 page 7, delate NWP. You only use numerical weather prediction one time.
*Response: Done*

Line 10 page 7, Lamb Weather Type Classification, delate technique.
*Response: Done*

In your paper you use autumn but in the figures you use fall. Please use one word (in UK you use autumn and in USA fall).
*Response: "Fall" word changed to "autumn" both in the Tables and Figures.*

Section 3.1. Delate Climatology, use another title.
*Response: We deleted "Climatology" and instead, "Spatial variation" was used*

Line 16 page 11. Add: As an example, on October...
*Response: Done*

Section 3.2. The parenthesis "(addition of...)" are confused, add the EPEs after each number.
*Response: We removed the numbers from the text in order to not confuse. Instead, the days were given in the Table 1.*

In sections 3.3 and 3.4. Make it clear that you are analyzing the main EPEs in section 3.2. And for this reason, you only analyze one EPE in section 3.4.
*Response: You are right. Therefore we changed the numbers of the subtitles. We used 3.2.1 and 3.2.2 instead of 3.3 and 3.4 sections.*

Lines 2 to 8, page 17. It seems a discussion, you can add some references to support your affirmations.
*Response: We added Baltacı et al. 2015, 2017 references to explain the sea-effect precipitation mechanism over Marmara in autumn season..*

[revised manuscript text omitted]